

# Energy balance of food in a detrito-bryophagous groundhopper (Orthoptera: Tetrigidae)

Kateřina Kuřavová[1], Jan Šipoš[2] and Petr Kočárek[1]

[1] Department of Biology and Ecology, University of Ostrava, Ostrava, Czech Republic
[2] Department of Zoology, Fisheries, Hydrobiology and Apiculture, Mendel University in Brno, Brno, Czech Republic

## ABSTRACT

Detritus (decaying organic matter) and phyllodes of mosses are two main components in the diet of groundhoppers (Orthoptera: Tetrigidae). We studied the energy balance of consumed food under laboratory conditions in the detrito-bryophagous groundhopper, *Tetrix subulata* (Linnaeus, 1758). The results indicated that the energy food budget of this detrito-bryophagous groundhopper was comparable to those of small herbivorous grasshoppers (Acrididae: Gomphocerinae, Melanoplinae), which have a similar energy food budget of approximately 800–1,100 J/g. *T. subulata* consumed four times more detritus than mosses, although both components provided similar amounts of energy (ca. 15–16 kJ/g). However, in contrast with detritus, moss fragments passed through the digestive tract without a distinct change in their mass or a loss in their energy value. We assume that moss may cause the longer retention of semifluid mass of partly digested food in the alimentary tract; hence, the digestion and efficiency of nutrient absorption from detritus could be more effective.

# INTRODUCTION

Optimal foraging theory predicts that a foraging organism will maximize its fitness by maximizing its net energy intake per unit of time (*Stephens & Krebs, 1986*) and will more often choose the available food components that yield the most calories for the effort it takes to locate, catch, or consume them (*Stephens & Krebs, 1986*). This theory explains natural foraging selection through quantitative models. Energy budgets are based upon the equation *Ingestion = (Assimilation = Production + Respiration) + Egestion* (*White & Watson, 1972*; *McEvoy, 1985*). Herbivores usually consume nutritionally rich food sources, and they select combinations of food that vary in quality and quantity (*Bernays, 1985*; *Bernays & Simpson, 1990*; *Simpson & Raubenheimer, 2012*). In general, little is known about the energy balances or associated food strategies of insects (especially in detritivores or bryovores).

Groundhoppers (Orthoptera: Tetrigidae) have a conservative feeding strategy of detrito-bryophagy (the consumption of detritus and lower plants) (e.g., *Verdcourt, 1947*; *Paranjape & Bhalerao, 1985*; *Hochkirch et al., 2000*; *Kočárek et al., 2011*; *Karpestam & Forsman, 2011*;

Corresponding author
Petr Kočárek, petr.kocarek@osu.cz

*Kuřavová & Kočárek, 2015*) that is conditioned by phylogenetic dietary conservatism (*Kuřavová et al., 2017a*). The proportions of consumed food components are very similar across different subfamilies and across species that occupy different habitats and live in different geographic regions, with detritus (soil particles with unidentified decomposed organic matter) comprising 80–90% of the diet, moss tissues comprising 15–12% of the diet, and residual matter (pollen grains, fungal hyphae, algae, mineral particles, and the body parts of various invertebrates) comprising 1–5% of the diet (*Kuřavová et al., 2017a*). Detritus is digested with higher efficiency (digestibility 91%) than moss tissue, which has a digestibility of approximately 60% (*Kuřavová et al., 2017a*). To date, the energy balance of the groundhopper foraging feeding strategy is unknown.

In this study, we focus on the following question: Is there any difference in the caloric values of the two dominant components of the groundhopper diet, and if so, does the selective utilization of these components correspond with the optimal foraging theory?

## METHODS

### Insect

*Tetrix subulata* is one of the most widespread groundhopper species in Europe (*Holst, 1986*). The body length of adults ranges from 10 to 14 mm, and females are usually larger than males (*Steenman, Lehmann & Lehmann, 2013*; *Steenman, Lehmann & Lehmann, 2015*; *Lehmann et al., 2018*). This species is active from March to the end of October in Central Europe (*Holst, 1986*, *Kočárek, Holuša & Vidlička, 2005*), when nymphs hatch in summer (August), molt to adulthood in autumn, hibernate and reproduce in spring. The adult season is split into the autumn dispersal-related cohort (*Lehmann et al., 2018*) and the reproducing spring cohort (*Steenman, Lehmann & Lehmann, 2015*). The groundhopper usually prefers damp places, and it is often found near rivers in moist habitats (*Baur, Baur & Roesti, 2006*). The diet of *T. subulata* includes detritus, mosses (e.g., genera *Brachythecium*, *Bryum*, *Calliergonella*), algae, and small amounts of other substrates (*Ingrisch & Kohler, 1998*; *Hochkirch et al., 2000*; *Kuřavová & Kočárek, 2017*).

### Experimental design

Randomly selected adults of *T. subulata* (brachypronotal or macroptonotal, brachypronotal dominated in a 4:1 ratio) of the spring cohort after the hibernation were collected by sweeping in flooded depressions of meadow near the city of Ostrava, Czech Republic (49°51′40.4″N, 18°11′19.5″E), from 20 Apr to 28 May 2014. Specimens were transported in plastic boxes to the laboratory at the Department of Biology and Ecology, University of Ostrava. The energy balance of food was evaluated using the gravimetric ingestion method (*Waldbauer, 1968*; *Kogan & Parra, 1981*; *McEvoy, 1985*). This experiment consists of the following parts: the acclimatization of specimens, food deprivation, feeding and calorimetric analysis of samples.

### Acclimatization

Specimens were acclimated in eight insectaria in laboratory conditions for three days (approximately 40 individuals per insectarium ($30 \times 15 \times 20$ cm) with a sex ratio of

1:1). The insectaria were ventilated by covering their tops with textile membranes with 1× one mm pores. Each insectarium contained a soil depth of seven cm (a mixture of detritus fragments), planted mosses, including *Brachythecium rutabulum* (Hedw.) B.S.G. and *Calliergonella cuspidata* (Hedw.) Loeske as well as the grass (*Festuca* spp.). The upper layer of the substrate was covered with 70% bare soil, 20% mosses and 10% grass. The soil layer in the insectaria was kept slightly wetted with water. Both substrate and all plants used in insectaria were collected from the same locality as the specimens.

## Food deprivation

The acclimated specimens were placed in plastic boxes ($25 \times 25 \times 25$ cm) to empty their digestive tracts. The boxes had double bottoms. The inner bottom was composed of a perforated textile membrane with 1× one mm pores, which allowed feces to pass through. Each box contained a wet inert fabric (cotton wool) to provide drinking water. Food deprivation lasted 24 h for each specimen. The sufficiency of food deprivation (i.e., the rate of food passage through the digestive tract) was tested experimentally in a microclimate chamber (Snijders Imago 500, Tilburg, The Netherlands) under laboratory conditions (temperature 25 °C, humidity 80%, photoperiod 12 h light and 12 h dark). The mean rate of food passage through the alimentary tract was approximately 5 h for detritus and approximately 7 h for the moss *C. cuspidata*. The last fecal pellet was recorded 22 h after the last consumption of detritus and moss.

## Feeding and feces collection

The starved specimens were weighed (analytical balances Kern EG 420; Balingen, Germany) to an accuracy of $10^{-3}$ g, and 20 individuals of the same sex were placed in boxes with one type of food: "feeding group with detritus" (a mixture of soil particles and decaying organic matter), "feeding group with *Calliergonella cuspidata*" (only phyllodes), and "feeding group with *Brachythecium rutabulum*" (only phyllodes). The food was collected from the same locality as specimens of *T. subulata* on 18 Apr 2014. Detritus was collected from the upper layer of decomposing phyto-organic matter found between the moss cushions. Phyllodes of the mosses *C. cuspidata* and *B. rutabulum* were manually collected. The diet was adjusted before it was served, the detritus and mosses were air-dried to a constant weight under laboratory conditions (temperature 23 °C, 48% humidity), and the dry matter was weighed (scale Sartorius AG, Göttingen, Germany) to an accuracy of $10^{-5}$ g due to the exact characteristics of the served food. The detritus and the mosses (in dried form) were analyzed for their carbon, hydrogen, nitrogen, sulfur, phosphorus, and silicon contents using the Leco CHN628 analyzer (Leco 628S, Saint Joseph, Michigan); the calorific values and the proportion of ash in the organic matrix were analyzed by the adiabatic calorimeter IKA C4000 (Staufen, Germany) (Table 1). Before serving, the detritus and the mosses were again wetted (moss was submerged in water for 20 min, and detritus was wetted with water to achieve a 2:1 ratio). Wet phyllodes of mosses and wet detritus were placed separately into boxes. All boxes, specimens and served food were controlled twice a day (in 12-h periods). Defecated feces fell individually down into a collection container (the lower of the container bottoms). The feces were collected continuously twice a day (in

Kuřavová et al. (2020), *PeerJ*, DOI 10.7717/peerj.9603
**Table 1** **The percentages of elements (C, H, N, S, P, S ), calorific values (CV$_{sf}$), energy values (E$_{sf}$) and proportions of ash matter (FA$_{sf}$) in foods served to Tetrix subulata at laboratory conditions.**[a] Bra-rut, moss *Brachythecium rutabulum*; Cal-cus, moss *Calliergonella cuspidata*.

| | C % | H % | N % | S % | P % | Si % | CV$_{sf}$ (cal/g ash-free dry wt) | E$_{sf}$ (kJ/g dry wt) | FA$_{sf}$ (%) |
|---|---|---|---|---|---|---|---|---|---|
| Detritus | $10.05 \pm 0.03$ | $1.61 \pm 0.02$ | $0.78 \pm 0.03$ | $0.19 \pm 0.03$ | $0.40 \pm 0.03$ | $56.75 \pm 0.25$ | $3839.85 \pm 41.74$ | $16.08 \pm 0.17$ | $70.89 \pm 1.22$ |
| Bra-rut | $36.53 \pm 0.36$ | $5.52 \pm 0.04$ | $1.22 \pm 0.03$ | $0.10 \pm 0.03$ | $3.07 \pm 0.09$ | $13.33 \pm 0.17$ | $3852.46 \pm 55.44$ | $16.13 \pm 0.23$ | $2.76 \pm 0.83$ |
| Cal-cus | $43.69 \pm 0.38$ | $5.86 \pm 0.05$ | $1.21 \pm 0.02$ | $0.10 \pm 0.03$ | $1.97 \pm 0.07$ | $14.49 \pm 0.18$ | $3599.08 \pm 5.81$ | $15.07 \pm 0.02$ | $2.50 \pm 0.06$ |

**Notes.**

[a]Three samples evaluated for each type of food.

12-h periods) during the 30 day feeding period. The collected feces were frozen at $-18\ °C$ (Beko freezer, CN 237231, Gaesti, Romania).

## Calorimetric analysis

The collected feces were analyzed with an adiabatic calorimeter (IKA C4000, Staufen, Germany), identical to the method used by *Hadley & Bliss (1964)*, *White (1978)* and *Köhler, Brodhun & Schäller (1987)*. Benzoic acid ($C_6H_5COOH$, 26 kJ/g) was used for calibration. The calorific value of a sample was expressed as calories per gram of ash-free sample. The ash residues were recorded for all samples.

A total of 240 individuals (20 males and 20 females in each feeding group with two replications) were used in this experiment. Acclimation, food deprivation, feeding, and fecal collection were conducted in a climate chamber (Snijders Imago 500, Tiburg, The Netherlands) with temperature held at 25 °C and humidity at 70% during the day (12 h of light) and 23 °C and 80%, respectively, during the night (12 h of dark). These conditions were constant over the experiment.

## Data analyses

The experiment was evaluated using the gravimetric method that relies on ingestion and egestion (*White & Watson, 1972*), where *Ingestion = Assimilation + Egestion*. The assimilated energy was determined by the difference between the initial (calorific value of ingested food component) and the final (calorific value of egested feces) dry matter using the following equations:

The calorific value of assimilated food ($CV_{diet}$) was calculated for the group of 20 individuals using the following formula:

$$CV_{diet} = CV_{digested} - CV_{feces},\ \ [J/g],$$

and $i$ is the type of food (detritus, moss species).

The real calorific value of assimilated food ($RCV_{diet}$) was calculated for each specimen using the following formula, and it is the calorific value that the specimen gains from food:

$$RCV_{diet\ i} = CV_{diet} - AD_{diet\ i},\ \ [J],$$

where $AD_{diet}$ is the approximate digestibility of food and $i$ is the type of food (detritus, moss). The approximate digestibility of food components in the groundhopper *T. subulata* was calculated by *Kuřavová & Kočárek (2017)* according to the following formula:

$$AD = (WCF - WF/WCF) \times 100,$$

where $WCF$ is the weight (mg) of the consumed food and $WF$ is the weight (mg) of the feces.

The energy food budget ($E_{fb}$) was calculated for each specimen using the following formula:

$$E_{fbi} = RCV_{dieti} \times 4.187,\ \ [J]$$

where *4.187* is the conversion factor used to convert the calorific value to joules, and $i$ is the type of food (detritus, moss species).

**Table 2 Mean weights of males and females belonging to three different feeding groups in *Tetrix subulata*: detritus, moss *Brachythecium rutabulum* (Bra-rut) and moss *Calliergonella cuspidata* (Cal-cus).** The mean weights of defecated feces (in dry matter) were collected for 30 days in laboratory conditions.[a]

| Feeding group | moss Bra-rut | | moss Cal-cus | | Detritus | |
|---|---|---|---|---|---|---|
| | Male | Female | Male | Female | Male | Female |
| Weight of specimens (mg) | 33.48 ± 7.15 | 71.58 ± 10.58 | 31.20 ± 2.45 | 71.18 ± 7.82 | 33.43 ± 2.05 | 71.70 ± 8.10 |
| Weight of feces (mg/spec./30days) | 11.81 ± 0.08 | 23.43 ± 1.63 | 10.40 ± 0.18 | 23.84 ± 0.53 | 18.36 ± 1.62 | 33.26 ± 0.43 |

Notes.
[a]Each feeding group had 20 specimens with two replicates.

The overall experimental design was a 2 × 3 factorial design (2 sexes and 3 levels of food source) with replicate measures at each level of explanatory variables. Sex, food source and their interaction were entered into the models as fixed effects, and replicates were entered as random effects. For data analysis, we used repeated-measures nonparametric ANOVA (ligned rank transformation ANOVA). It is a robust statistical tool for the analysis of multiple factorial designs with non-normal residuals. Before using ANOVA itself the data were transformed by the "art" function (ARTool package) (*Wobbrock et al., 2011*). This function first aligns the data for each effect (main or interaction) and then assigns averaged ranks (*Mansouri, 1998*). The post hoc comparison of the main effect for food source was conducted by the "emmeans" package with Bonferroni corrected *p*-values (*Russell, 2019*). All analyses were performed using the statistical software R (Ver. 3.1.3, Vienna, Austria) (*R Core Team, 2015*). The level of probability was considered significant at a *P*-value < 0.05.

## RESULTS

Females weighed an average of 71.49 ± 8.83 mg and were therefore nearly twice as heavy as males (32.70 ± 3.88 mg) ($df = 1$, $F = 686.70$, $p < 0.01$, Table 2). The weights of males and females did not differ significantly among feeding groups consuming different types of food ($F_{2,3.9} = 4.46$, $p = 0.096$, Table 2). The fecal weights significantly differed between males and females ($F_{1,6.2} = 19.08$, $p < 0.01$), and females had approximately 13.32 ± 0.53 mg (ca 50%) heavier feces than males.

The calorific values of served food differed significantly from each other ($F_{2,4} = 7$, $p = 0.049$, Table 1). The Tukey HSD test confirmed that the calorific values of detritus and *Brachythecium rutabulum*, and *Calliergonella cuspidata* mosses were similar, but the calorific values of *B. rutabulum* moss were slightly different from those of *C. cuspidata* moss (Table 3). Ash matter significantly differed among the types of food served ($F_{2,4} = 11.46$, $p = 0.022$, Table 1). Detritus contained more ash matter than both mosses. The Tukey HSD test confirmed that the ash matter slightly differed among the types of served food (Table 3) but not between the two served mosses. The energy food budgets in individual feeding groups and this parameter differed significantly between the type of served food ($F_{2,8} = 18.51$, $p < 0.001$) and between males and females ($F_{1,4} = 37.39$, $p < 0.01$) (Table 4).

The energy food budgets of *T. subulata* are comparable to the energy food budgets of small herbivorous grasshoppers from the family Acrididae (Acridinae: *Chorthippus biguttulus*, *Gomphocerippus rufus*, and *Pseudochorthippus parallelus*; Melanoplinae:

**Table 3** Tukey multiple comparisons of calorific values (value before the slash) ($CV_{sf}$) and ash matter (value after the slash) ($FA_{sf}$) in served food. moss Bra-rut, *Brachythecium rutabulum*; moss Cal-cus, *Calliergonella cuspidata*; Efb, energy food budgets in feeding groups of groundhopper Tetrix subulata. The values represent the honest significant difference (*P*-value).[a]

| Type of food | $CV_{sf}$ | $FA_{sf}$ | $E_{fb}$ |
|---|---|---|---|
| Moss Bra-rut –Detritus | 0.71 (0.77) | −2.83 (0.07) | −3.97 (<0.01) |
| Moss Cal-cus –Detritus | −2.83 (0.07) | −3.54 (0.03) | −5.56 (<0.01) |
| Moss Bra-rut –Moss Cal-cus | 3.54 (0.03) | 0.71 (0.77) | 1.59 (0.29) |

**Notes.**
[a] Each feeding group had 20 specimens with two replicates (for a total of 240 individuals).

**Table 4** Summary of caloric values and proportions of ash matter in assimilated food and defecated faces for males and females in *Tetrix subulata*. $CV_{df}$, calorific values of defecated feces per feeding group; $FA_{df}$, proportions of ash matter in feces per feeding group, $CV_{af}$, calorific values of assimilated food per specimen collected for 30 days; $RCV_{af}$, real calorific values of assimilated food per specimen collected for 30 days; $E_{fb}$, energy food budget ($E_{fb}$) per specimen in feeding groups of *Tetrix subulata*. The feeding groups consumed three types of food: moss *Brachythecium rutabulum* (Bra-rut), moss *Calliergonella cuspidata* (Cal-cus), and detritus. The values are mean ± standard error.[a]

| Feeding group | *moss Bra-rut* | | *moss Cal-cus* | | *Detritus* | |
|---|---|---|---|---|---|---|
| | **Male** | **Female** | **Male** | **Female** | **Male** | **Female** |
| $CV_{df}$ (cal/g) per group/30 days | 3217.21 ± 344.38 | 3752.62 ± 32.48 | 3164.86 ± 426.33 | 3572.08 ± 32.75 | 280.67 ± 29.21 | 862.17 ± 3.66 |
| $FA_{df}$ (%) per group/30 days | 4.71 ± 2.09 | 0.53 ± 0.09 | 6.30 ± 1.95 | 2.10 ± 0.47 | 85.86 ± 1.45 | 55.50 ± 0.12 |
| $CV_{af}$ (cal/g/spec./30 days) | 31.76 ± 14.45 | 4.99 ± 1.15 | 21.71 ± 21.03 | 1.35 ± 1.35 | 177.96 ± 0.63 | 148.88 ± 1.90 |
| $RCV_{af}$ (cal/spec./30 days) | 5.27 ± 2.40 | 1.84 ± 0.42 | 3.17 ± 3.07 | 0.51 ± 0.51 | 290.82 ± 1.02 | 477.28 ± 6.10 |
| $E_{fb}$ (J/spec./day) | 0.73 ± 0.33 | 0.26 ± 0.06 | 0.44 ± 0.43 | 0.07 ± 0.07 | 40.59 ± 0.14 | 66.61 ± 0.85 |

[a] There were three replicates for each type of food. There were 20 males and females in each group with two replicates (for a total of 240 individuals).

*Melanoplus femurrubrum*, and *M. sanguinipes*), with similar energy food budgets of approximately 800–1,100 J/g (Fig. 1).

The groundhopper *T. subulata* consumes four times more detritus than mosses, although both food components provide a similar amount of energy (Table 1). Females obtain more energy from food than males. Feces contain more moss fragments than other waste products in a proportion of 4.5:1 (Table 4). Assimilated energy of served food in males and females was significantly different ($F_{2,8} = 71.09$, $p < 0.001$) in that males obtain more energy from mosses than females, but females obtain more energy from detritus (Fig. 2). The Tukey HSD test confirmed that the energy budgets of specimens differed between the detritus and moss feeding groups (Table 3, Fig. 3), but not between the two moss feeding groups.

## DISCUSSION

Based on the gravimetric method that relies on ingestion, we confirmed that energy food budgets differ between two dominant food components in the detrito-bryophagous groundhopper *Tetrix subulata*. Decaying organic matter (detritus) is assimilated more effectively, has higher digestibility, and provides more energy than moss tissues (Fig. 3).

The dietary preferences of groundhoppers are relatively well known; the main component of their diet is detritus, and minor components include a mixture of moss species (*Kočárek et al., 2011*, *Kuřavová & Kočárek, 2015*; *Kuřavová & Kočárek, 2017*). Groundhoppers
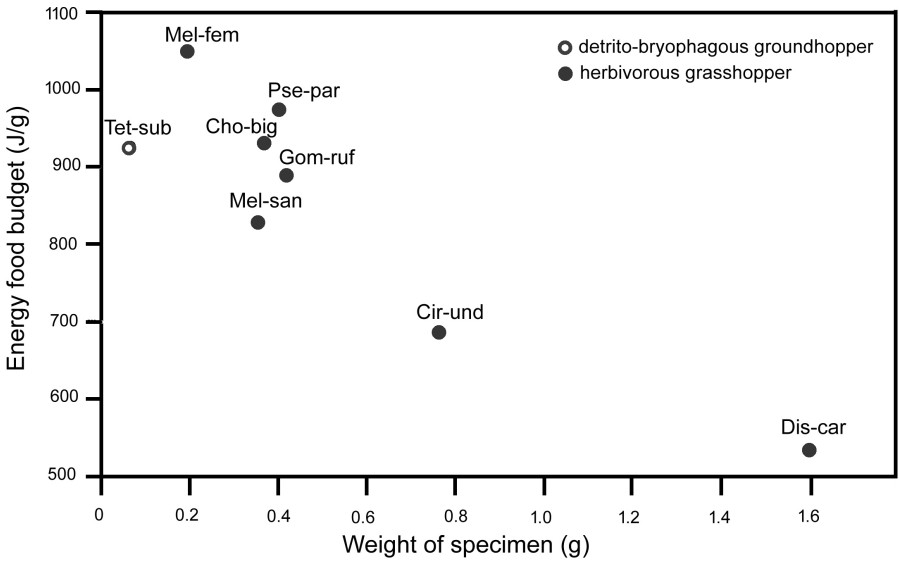

**Figure 1** Energy food budgets (J/g) in adult females of various Caelifera species determined through the gravimetric ingestion method according to *Belovsky (1986)*, *Köhler, Brodhun & Schäller (1987)* and our result. **Cho-big**, *Chorthippus biguttulus* (Linnaeus, 1758); **Cir-und**, *Circotettix undulatus* (Thomas, 1872); **Dis-car**, *Dissosteira carolina* (Linnaeus, 1758); **Gom-ruf**, *Gomphocerippus rufus* (Linnaeus, 1758); **Mel-fem**, *Melanoplus femurrubrum* (De Geer, 1773); **Mel-san**, *Melanoplus sanguinipes* (Fabricius, 1798); **Pse-par**, *Pseudochorthippus parallelus* (Zetterstedt, 1821); **Tet-sub**, *Tetrix subulata* (Linneus, 1758).

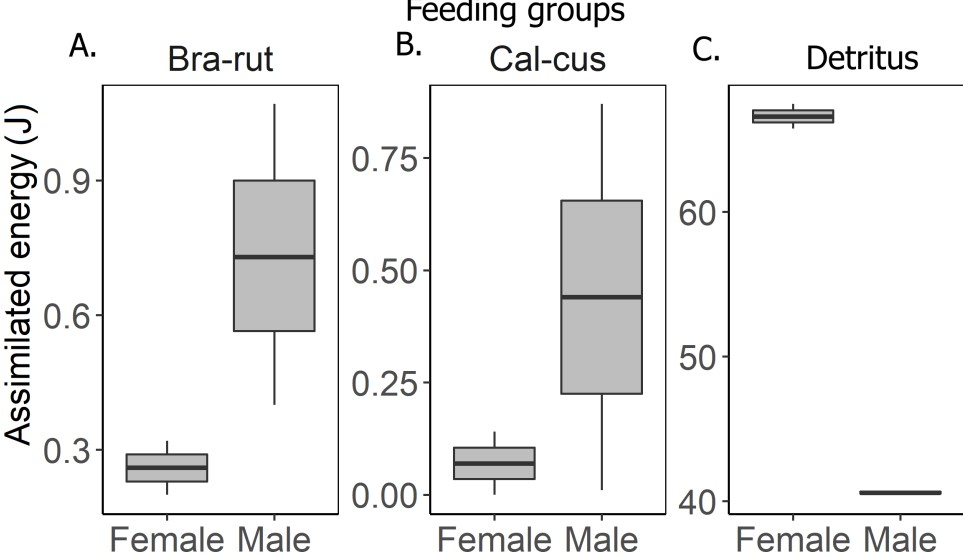

**Figure 2** Assimilated energy in males and females of various feeding groups. (A) moss *Brachythecium rutabulum* (Bra-rut), (B) moss *Calliergonella cuspidata* (Cal-cus), (C) detritus.

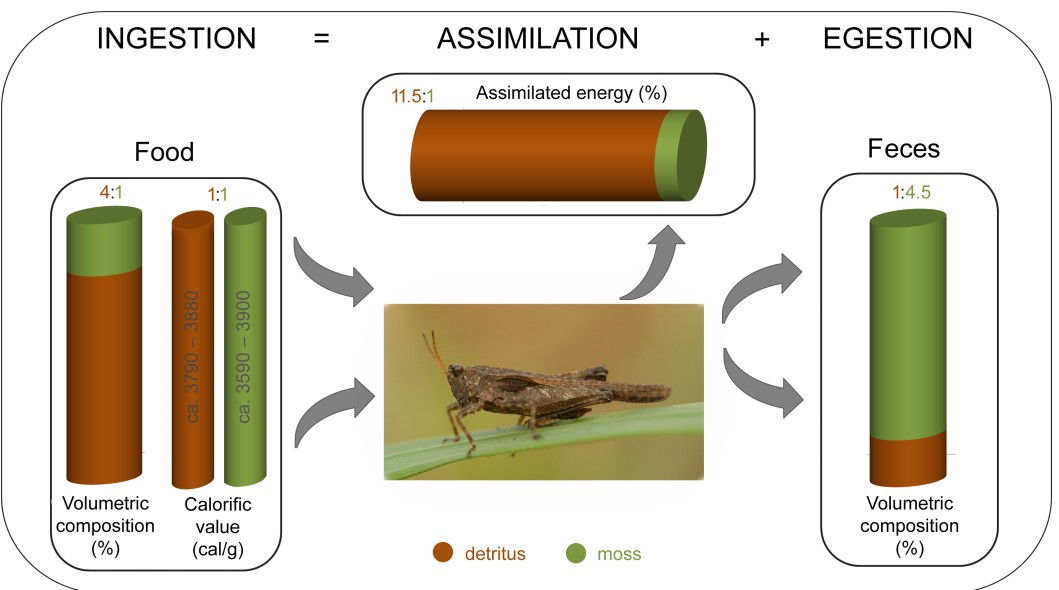

**Figure 3 Food balance in detrito-bryophagous groundhopper *Tetrix subulata* (Orthoptera, Tetrigidae).** The food box shows the rate of detritus and moss consumption in studied groundhoppers (determined through gut content analysis according to *Kuřavová et al. (2017a)* and *Kuřavová & Kočárek (2017)*, and the calorific values of served detritus and moss tissues measured using calorimeter method. The assimilation box shows the proportion of assimilated energy (%) (assimilation means respiration and production of energy). The feces box shows the proportion of defecated feces (determined through the gut content analysis according to *Kuřavová & Kočárek (2017)*.

frequently consume moss species that are dominant at each locality, but some species-specific preferences in moss consumption have been observed (*Kuřavová et al., 2017b*). Both basic components of groundhopper diets (detritus and moss) differ in chemical composition (*Frankland, 1974*; *Rice, 1982*; *Enríquez, Duarte & Sand-Jensen, 1993*; *Asakawa, 1995*; *Asakawa, 2007*; *Maksimova et al., 2013*). Conducted analyses show that detritus has a higher percentage of silicon, while moss is richer in other elements (particularly carbon, phosphor, Table 1). The results of our elemental analysis of mosses are comparable with the ranges of basic elements found in other bryophytes (*Maksimova et al., 2013*). In terms of energy food richness, both food components were balanced (i.e., provided approximately 15–16 kJ/g), but detritus seemed to contain more easily digestible compounds than moss (*Kuřavová & Kočárek, 2017*).

Insect energetics, qualitative nutritional requirements of insects, consumption rates and energy balances in insects have all been studied (see *Wiegert & Petersen, 1983*; *McEvoy, 1985* for reviews), and the energy budgets have also been analyzed in some herbivorous grasshoppers (e.g., *Nagy, 1952*; *Duke & Crossley, 1975*; *White, 1978*; *Belovsky, 1986*; *Köhler, Brodhun & Schäller, 1987*). The contents of the alimentary tracts often included a mixture of grass species in the studied grasshoppers, e.g., *Lolium perenne* L., *Poa pratensis* L., *Festuca rubra* L. and *Dactylis glomerata* L. We analyzed the energy food budget of a detrito-bryophagous groundhopper using the gravimetric method proposed in the abovementioned studies (*Belovsky, 1986*; *Köhler, Brodhun & Schäller, 1987*). We can

compare the energy food budgets across used Orthoptera species, and the results indicated that the energy food budgets are comparable in small herbivorous grasshoppers of similar weight (see Fig. 1) .

The theory of optimal foraging strategy explains multidimensional feeding selection in various animals (*Stephens & Krebs, 1986*; *Raubenheimer & Simpson, 1993*; *Sinervo, 1997*), whereas herbivorous insects require food with a mixture of nutrients to sustain growth, development and reproduction, and they must regulate their nutrient intake (*Behmer, 2009*). Consumers always try to obtain an optimal balance of food components over time (*Berner, Blanckenhorn & Körner, 2005*; *Simpson, 1990*; *Simpson et al., 2002*). We found that *T. subulata* consumes significantly greater amounts of detritus than mosses, although both food components provide a similar amount of energy (ca. 15–16 kJ/g) (Fig. 3). This variance has important implications for assimilation and energy yield.

Based upon our results, detritus is a better energy food source for groundhoppers than mosses in proportion 11.5:1 (Fig. 3), whereas moss tissues pass through the alimentary tract without providing significant energy benefits for specimens. During our previous studies we tested two working hypotheses: (1) moss tissues are a significant source of water in dry season/day periods, and (2) groundhoppers might consume mosses to obtain cryoprotectants (*Cornelissen et al., 2007*) in the case of an autumn cohort. *Kuřavová et al. (2017b)* studied whether groundhoppers consumed mosses to obtain water by comparing of the food composition at two sites that differed considerably in water availability (humid vs. dry microhabitat). The results suggest that the studied species *T. tenuicornis* and *T. ceperoi* predominantly consumed the available mosses, i.e., the most frequently consumed mosses were the dominant species at each site. Regardless, some desiccation-tolerant (and concurrently nutritionally rich) moss species seemed to be more consumed at the dry versus the humid site. The second hypothesis was rejected based on the finding that groundhoppers consumed more mosses in spring and summer than in autumn before hibernation (*Kuřavová & Kočárek, 2015*).

The most likely hypothesis, which could explain the regular consumption of moss, seems to be that moss fragments facilitate a longer retention time of chyme in the alimentary tract, improving digestion and efficiency of nutrient absorption. Therefore, moss tissues may perform the same function as dietary fiber in omnivorous vertebrates (*Truswell, 1993*). Evidence for this claim is that the passage of detritus through the alimentary tract is faster than that of moss (on average 5 h vs. 7 h). The rate of the passage of different foods through the alimentary tract is the subject of ongoing experiments (K. Kuřavová, 2020, unpublished data). Groundhoppers are a phylogenetically ancient group of orthopterans that exhibit conservative feeding strategy (*Kuřavová et al., 2017a*), which could be associated with the absence of an enzymatic apparatus necessary for the digestion of some nutrients (esp. polysaccharides) of vascular plants. Basic nutrients (saccharides, proteins) are accessible in partially digested form in detritus (oligosaccharides, oligopeptides); thus, they are easier for groundhoppers to digest and can compensate for the absence (or low effectiveness) of their own specific enzymes. *Kuřavová, Hajduková & Kočárek (2014)* found a high level of mechanical wearing of mandibles as a result of feeding in *T. tenuicornis*. Less sclerotized

and easily abradable cuticle may be one of the the reasons why groundhoppers avoid feeding on silica-rich higher plants.

In conclusion, we evaluated the energy balance of food in a detrito-bryophagous groundhopper under laboratory conditions. Detritus is consumed and digested more efficiently and is the most significant energy source in the groundhopper diet. Moss tissues pass through the digestive tract in almost unchanged form; therefore, we conclude that mosses are unimportant sources of energy for groundhoppers. Moss fragments may cause the longer retention of chyme in the alimentary tract; hence, the digestion and efficiency of nutrient absorption could be more effective.

## ACKNOWLEDGEMENTS

The authors thank chemist Boleslav Taraba (CZE) and technologist Jiří Fiedor (CZE) for providing laboratory equipment during the experiment. We thank Carlos Sperber, Axel Hochkirch and a third anonymous reviewer for very helpful comments that improved the manuscript.

### Funding

This research was supported by project CZ.1.05/2.1.00/19.0388 of EU structural funding Operational Programme Research and Development for Innovation, project LO1208 of the National Feasibility Programme I of the Czech Republic, and by an Institutional Research Support grants from the University of Ostrava (SGS21/PřF/2013, SGS24/PřF/2014 and SGS2/PřF/2015). The funders had no role in study design, data collection and analysis, decision to publish, or preparation of the manuscript.

### Grant Disclosures

The following grant information was disclosed by the authors:
EU structural funding Operational Programme Research and Development for Innovation: CZ.1.05/2.1.00/19.0388.
National Feasibility Programme I of the Czech Republic: Project LO1208.
Institutional Research Support grants from the University of Ostrava: (SGS21/PřF/2013, SGS24/PřF/2014, SGS2/PřF/2015.

### Competing Interests

The authors declare there are no competing interests.

### Author Contributions

- Kateřina Kuřavová conceived and designed the experiments, performed the experiments, analyzed the data, prepared figures and/or tables, authored or reviewed drafts of the paper, and approved the final draft.
- Jan Šipoš analyzed the data, prepared figures and/or tables, and approved the final draft.
- Petr Kočárek conceived and designed the experiments, authored or reviewed drafts of the paper, and approved the final draft.

## Data Availability

The raw measurements are available in a Supplementary File.

## Supplemental Information

Supplemental information for this article can be found online at http://dx.doi.org/10.7717/peerj.9603#supplemental-information.

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
