# Peer review of "Energy balance of food in a detrito-bryophagous groundhopper (Orthoptera: Tetrigidae)"

_PeerJ, doi:10.7717/peerj.9603_

## Round 0.1 · original submission · Major Revisions

I received reports from three reviewers and all agree that despite the general content of the study is fine, there are also very strong linguistic issues in addition to statistical problems that need to be addressed. All the reviewers have made many suggestions to help rewriting the manuscript, but I also suggest to search for the help of a native English speaker colleague. I encourage you to take these suggestions into account before submitting a revised manuscript.

·

Basic reporting

1. First check on figures: ok
2. First check on tables:

Table 3 – I have not understood Table 3: apparently the authors opted to organize a three-way table (actually 3 x 2 = six way, i.e., three types of food, multiplied by two sexes), in a 4 by 2 table (dis-considering the two variables “Weights of defecated feces” and “Energy food budgets”, which led the table to be pretty confusing and difficult to interpret. I would suggest a more straightforward organization, with the type of food organized as three distinct columns, sex as two lines, in two horizontal groups of lines, one pair (males and females) for “Weights of defecated feces” and another pair (males and females) for “Energy food budgets”. If I misunderstood the table, I am terribly sorry, but this denotes that it has a rather awkward presentation.

Table 4 – Here again, the author opted for a the same awkward organization used in Table 3. Please, I would ask the authors to consider remodeling of these tables’ organization.


Table 5 – Here the table’s organization is much better!


Tables were excessive and confusing. I recommend that the authors substitute most table by figures, showing more clearly what treatment levels (Food items) were different from each other. It was not possible to understand clearly the results of the a multiple average tests (Tukey multiple comparisons). I would suggest that, in state of this approach, the authors could use a more straightforward contrast analysis (see, for example, Crawley (2005): chap. 11), testing explicit hypotheses. They would probably also be a posteriori, but the presentation and interpretation of the results would be easier to follow, and thus, to interpret.

The authors presented two figures, which presented quite interesting and relevant results, but awkwardly they cited them only in the discussion section. I would recommend them to include the citation and presentation of these figures in the results section.

I would suggest that, after transforming all or most of the tables in figures, the authors should reword the manuscript, so as to make it more clear. Furthermore, I would suggest them to ask for a thorough native English speaker review, so as to facilitate comprehension by the reader.

I considered the experiment and the results extremely relevant and interesting, but the wording and presentation of the manuscript were very confusing and difficult to follow. At the same time, the text was repetitive. Worst of all, the results were not clearly presented, neither in the text, nor, much worse, in the tables.

I would encourage the authors to invest in this manuscript, through a more straightforward statistical analysis and presentation of the results, and through a rewording of the text, so as to make it more direct and clear.
I would like to emphasize that the manuscript is scientifically valid and suitable, but it requires further improvement in the presentation and wording, so as to be more clear to the readers.
Raw data can be opened, but are not in English, but Czech. This is not a critical problem, as far as the interested reader can use automatic translator sites, which work reasonably well, especially considering there are few and rather simple words, whose meaning can be checked and eventual translation errors can be pinpointed.

English language is not clear, difficult to understand and rather repetitive. I would recommend that the authors ask for a native English revision.

Author comments are impressively better written that the manuscript. I would be happy to find such wording along the manuscript itself.

Experimental design

I missed an explicit presentation of tested hypotheses, and an explicit connection between the research questions and the methodology. It was difficult to understand which were the so-called feeding groups (there were three diets, as far as I understood – detritus and two moss species): the authors mentioned 240 individual groundhoppers, being 20 males and 20 females in each feeding group, which sums to 120 individuals, not 240. I certainly missed something here. This seemingly minor doubt shows the difficulties I had to understand the experimental design. By the way, the authors seem to have confounded experimental design, related to replications, factors and factor levels (treatments, sex) with the description of the laboratory methodology itself.

Validity of the findings

The findings are valid and relevant. Very little is known on the diet and energy balance of feeding in orthopterans (although there were the publications cited by the authors, including themselves). Such information is of utmost importance to understand the ecological processes involved in this insects’ survival, besides providing information that can be extrapolated, certainly with caution, to other related insects, for example other orthopterans that also inhabit the same microhabitats.

Additional comments

I would recommend the acceptance of this manuscript, but not in the present form. The manuscript necessitates thorough rewording, reduction, reorganization and probably substitution of most tables by figures, inclusion of the figures in the results section – and consequently, inclusion of the reasoning, hypotheses and methodologies for the comparisons, in the appropriate sections, as well as a thorough English revision, so as to facilitate comprehension. Nevertheless I would instate the authors to invest this effort to ameliorate the manuscript, and thus, provide an important contribution to science on orthopterans and insect ecology.

·

Basic reporting

The authors have studied the energy balance of Tetrix subulata in the lab. The results are very interesting and worth publishing. However, there are numerous linguistic issues with the manuscript. I have made some suggestions for changes below, but it would be great if could be checked by a native-speaker. There are only a few sentences, which do not need corrections. I found the discussion rather weak. In most cases, only the own studies are cited, while the content of similar studies on insects is not well integrated in the text (just a list is provided).

Experimental design

The experimental design of the study was good and the research question was interesting. The methods are described with sufficient detail.

Validity of the findings

The findings are quite novel and provide new insight into the feeding ecology of groundhoppers. The underlying data are robust and statistically sound. The conclusions are well stated and linked to the original research questions (although there are a couple of redundancies in the dicussion). The authors also present some alternative hypotheses regarding the value of bryophyte feeding in this species.

Additional comments

Specific comments:
L 23: Either it should be “budgets of detrito-bryovorous grasshoppers” or “budget of the detrito-bryovorous grasshopper”.
L 23: Not clear what is meant here with “comparable with the small herbivorous grasshoppers” (is this related to other grasshopper species?)
L 26: Add “The” before “groundhopper”.
L 28: Replace “passes” by “pass”.
L 30: Replace “are unimportant source” by “are not an important source”.
L 30-31: Not clear what is meant with “whereby their significance in the diet is different”.
L 35: Replace “to make sense of” by “understand”.
L 36: Can you replace “the relevant inputs” with something more precise?
L 42-43: Replace “and they combine various choose in quality or quantity” by “, which may vary substantially in quality or quantity”.
L 44-45: Better start this sentence with “However,” and add a reference at the end of it.
L 46: Replace “show” by “have a”.
L 49: Replace “conditioned by” by “as a result of”.
L 49: Add “The” before “proportions”.
L 50: Replace “is” by “are”.
L 50: Replace “occupied” by “occupying”.
L 52: Replace “of moss tissues” by “moss tissue”.
L 53: Replace “of residual matters” by “residual matter”.
L 55: Replace “moss tissues” by “moss tissue”.
L 67: Replace “near rivers on the moist habitats (wet meadows)” by “in moist habitats, such as wet meadows, river banks, ephemeral ponds etc.”
L 73: Replace “of meadow” by “of a meadow”.
L 76: Replace “by” by “using the”.
L 80: Replace “had been” by “were”.
L 82: Replace “had been” by “were”.
L 82: Add “a” before “textile membrane”.
L 83: Replace “contained the soil to 7 cm depth” by “contained a 7 cm layer of soil”
L 85: Replace “and grass” by “as well as the grass”.
L 85: Add “the” before “substrate”.
L 86: Remove “covered” before “by grass”.
L 87: Add “The” before “substrate” and before “insectaria”.
L 92: I don’t know what an “inert fabric” is.
L 97: Remove “moss”.
L 102: Remove “moss”.
L 103: Remove “moss”.
L 119: Add “a” before “30 days”.
L 120: Replace “freeze” by “frozen”.
L 124: Add “The” before “calorific value”.
L 124: Add “each” before the first “sample”.
L 124: Replace “sample of ash-free” by “ash-free sample”.
L 127: Replace “collections” by “collection”.
L 139: Add “each” before “specimen”.
L 144: Add “each” before “specimen”.
L 148: Add “the” before “groundhopper”.
L 150: Add “each” before “specimen”.
L 152: Add “the” before “conversion factor”.
L 152: Replace “transfer” by “converting”.
L 152: Add “the” before “type of food”.
L 156: Add “the” before “weight”.
L 157: Add “each” before “feeding group”.
L 158: This sentence needs to be completely re-written.
L 162: Change to “We used Tukey’s HSD test to compare differences…”.
L 168: This sentence is somewhat unnecessary. Better just explain the differences as in the next sentence and refer to the table at the end of this sentence.
L 171: Replace “not differed” by “did not differ significantly”.
L 171: Replace “consumed” by “consuming”.
L 172: Delete the sentence “Mean weights…” and just add the reference to table 2 in the parentheses of the next sentence.
L 173: Replace “significantly differed” by “differed significantly”.
L 174: Replace “have” by “had”.
L 175: Replace “consumed” by “consuming” (or just “fed on different diets” instead of “consumed different type of food”).
L 176: Add “The” before “Tukey”.
L 177: Add “the” before “weight”.
L 177: Delete “defaecated” (it should be clear that “faeces” is defecated).
L 177: Replace “between served food” by “between different food types”
L 178: Replace “but no between two served moss species” by “but not between moss species”.
L 178-180: This sentence is not clear. Do the numbers refer to the different sexes or the different food types?
L 181-182: Replace this sentence by “The calorific value of the different food types (Table 1) differed significantly (df = 2….).
L 182: Add “The” before “Tukey”.
L 182: Add “the” before “calorific”.
L 183: Add “the” before “moss” (2 times).
L 184: Add “the” before “served food”
L 184: Delete “shows Table 1 and its matter” and add “Table 1” to the parentheses at the end of the sentence.
L 185: Change “significantly differed” to “differed significantly”.
L 186: Add “The” before “Tukey HSD test”.
L 187-188: Delete this sentence.
L 188: Add “The” before “Energy”.
L 189: Delete “show Tables 3, 5 and this parameter” (add the reference to the tables in parentheses at the end of the sentence).
L 189: Change “significantly differed” to “differed significantly”.
L 191: Add “The” before “Tukey HSD test”.
L 195: Replace “Base on” by “Based upon”.
L 195: Replace “are different” by “differ”
L 196: Add “the” before “detrito-bryovorous”.
L 198: Change “the moss tissues” to “moss tissue”.
L 198: Change “Mosses are unimportant” to “Mosses appear not to be important”.
L 199: What is meant with “their portion in the food seems different”. Could mosses have any other benefit for digestion (e.g. comparable to dietary fibres?).
L 204: Replace “consumed” by “consume”.
L 205: Delete “the” at the end of the line.
L 208-209: Delete the sentence “We provided…” (the reader should know this by now).
L 209-210: Rewrite this sentence as follows: “Our analyses show that detritus has a higher percentage of silicon, while moss is richer in other elements (particularly carbon, phosphor, Table 1).”
L 211: Add “The” before “Results”.
L 213: Delete “the” before “both food components”.
L 215-218: Rather than just listing this high number of references, it would be good to compare your results with those from these studies.
L 220-221: Not clear what this sentence refers to. I assume it refers to the references given before, but this may not be clear to the reader.
L 222-223: Delete this sentence (the reader should know this meanwhile).
L 223-224: Rather than stating that you “can compare” the results, just do it!
L 227: Replace “feeding” by “feed upon”.
L 227: Replace “to growth” by “for growth”.
L 228: Add “The” before “Theory”.
L 231: What is meant with “better balanced food components per time”?
L 232: Add “The” before “Groundhopper”.
L 232: Replace “bigger amount” by “larger amounts”.
L 233: Delete “the” before “both”.
L 233: Add “a” before “similar”.
L 234-235: Replace “in detritus, than in moss tissues” by “of detritus compared to moss tissue”.
L 235-236: Not clear what this sentence should mean.
L 236: Replace “We recorded, that” by “Based upon our results”
L 236: Add “a” before “better”.
L 236: Delete the comma.
L 237: Add “the” before “alimentary tract”.
L 238: Delete “Defaecated”.
L 238: Replace “, than other waste products” by “and contain only few other waste products”.
L 239-241: Not clear why this corresponds with the optimal foraging strategy. Why do they eat mosses at all?
L 241: Better suggest some alternative functions based upon the literature or own ideas.
L 244: Replace “moss tissues are” by “moss tissue is”.
L 244: Why “Another”? You didn’t present any other hypotheses before! Rewrite this sentence as follows: “It was hypothesized before that groundhoppers might consume mosses to obtain cryoprotectants (Cornelissen et al., 2007) before hibernation, but …
L 245: Replace “was possibility” by “is the possibility”.
L 248: Replace “the overwintering” by “hibernation”.
L 248-252: I think the sentences “We can summarize” to “unanswered” can be deleted. The authors should be aware of these findings after reading the text.
L 252: Replace “One explanation” by “Another explanation”.
L 253: Replace “can be matter for longer retention” by “facilitate a longer retention time”
L 253: Add “the” before “alimentary”.
L 254: Replace “whereby the digestions and efficiency of nutrient absorption could be more effective in these circumstances” by “improving digestion and efficiency of nutrient absorption”.
L 255: Replace “can” by “may” (good to see this hypothesis here, see my comment above).
L 257: Replace “It changing” by “It influences”.
L 258: Add “an” before “ancient group”.
L 258: Delete “own”.
L 259: Add “the” before “gut”
L 259: Replace “may be unable” by “may not be adapted“.
L 261-262: Replace this part by “which may compensate the lack of enzymes or an own gut flora of the groundhoppers”.
L 262: Add “probably” before “accessible”
L 264-272: This part is largely redundant and should be deleted. Instead, you might enter a short conclusion particularly focussing on what steps should be done next (e.g. analysing mixed diets).
Figure Heading to Figure 2: add “the” before “calorific values” and replace “by calorimeter method” by “using the calorimeter method”.

Reviewer 3 ·

Basic reporting

Dear authors, dear Editor,

I like the chosen research topic. Unfortunately, the manuscript needs a major revision before it can be fully evaluated.
First, even if I am not a native speaker myself, I need to mention that the English is not very good. I was in some parts even unable to understand the meaning (e.g. lines 30-31). I encourage the authors to ask a native speaker to improve the writing.
Otherwise, I see some reasons in the structure and the presentation that a serious and very detailed revision can be done to improve this manuscript, see my suggestions in the next two sections and specifically in the attached PDF file.

Experimental design

The experimental design seems to be quite reasonable, however necessary information are omitted:
I) did you use only individuals of one out of two wing morphs?
II) please specify the types of balances
III) give at least a brief explanation of the elementary analysis
It is unclear to me why you used 240 individuals? You had six groups (3 different diets by 2 sexes) each with 20 animals, accounting for 120 individuals. You mentioned a replicate at least in the tables? This would explain the 240 individuals, but in that case, you need to explain this. I even thing that you need to include a factor replicate into your statistics, see also the next section.

Validity of the findings

It seems that most data have been provided, however the tables are presented in Czech! This does not allow me to check their validity.

More importantly, the statistics seems to be wrong: if three different feeding types were tested (moos species 1, moss species 2, detritus) in two different sexes (males, females) the statistical analysis is a three by two factor analysis, e.g. using ANOVA's. Instead of this, the authors split the statistics and test between sexes, see lines 168-169 for body masses, lines 173-174 for faeces, followed for a test between diets, see lines 170-172 for body masses and lines 174 to 176 for faeces. This is not appropriate and must be done in a single analysis.

We need to await the revision, before a full assessment of the validity of the presentation and the Interpretation can be conducted.

Annotated reviews are not available for download in order to protect the identity of reviewers who chose to remain anonymous.

---

## Round 0.2 · Major Revisions

One of the previous reviewers accepted to make a second turn of review for the manuscript but raised once again major issues about the writting style which is too poor and about presentation of the data which is unclear. Without significant amelioration on both of these topics, it will not be possible to accept the paper for publication.

Reviewer 3 ·

Basic reporting

Dear authors, dear Editor,

I still like the chosen research topic of this revised manuscript. However, after some hours of critical proofreading and commenting I still see severe problems. The manuscript needs a very major revision before it can be fully evaluated.
The English has increased since the initial submission, but some parts are still hard to understand. This starts with mixed usage of British or American English (in case of faeces or feces). I am unsure whether these problems are mere language problems or are indeed problems with the logic. However, the authors need to improve the representation in any part of the manuscript.
I have made many suggestions in the attached PDF file.

To pick just a few problems:
I) The writing is unspecific, not to the point. I have tried to make a huge amount of suggestions directly in the text.
Ia) To illustrate a very bad style:
The authors write in Line 189: “Calorific values of served food are shown in Table 1….”.
In a scientific text you never show something, you describe facts. The text should be specific, and the convention is that the text stands for its own and must be understandable without looking into a figure or a table. The same applies to figures or tables, they must be understandable without back and forth checking the text. This is no language problem but a poor style of presentation. The same issue in lines 186ff, 193ff

Ib) Mass and weigh are not used properly, see line 169.
Comment: Physically you weigh something to measure its mass

Ic) The season of the species does not continue from March to October as written in line 72. Comment: This is not exactly true: nymphs hatch in summer (August), molt to adulthood in autumn, hibernate and reproduce in spring. So the adult season is split into the autumn dispersal related cohort (Lehmann et al. 2018) and the reproducing spring cohort (Steenman et al. 2015).

Experimental design

II) There are problems with the data presentation:
IIa) The presentation is very hard to follow, sometimes data are presented in Joule (the SI unit), sometimes in calories, sometimes units are missing entirely (see Figure 2).
IIb) I was not able to find out, whether data are presented as group means or adjusted to individual data.
IIc) Data are sometimes separated for sex (see Table 4) or the same data lumped across sexes as in Table 1. The authors need to be consistent across the manuscript.

III) The experimental still lacks some necessary information:
IIIa) you use individuals of both wing morphs, in which composition?
IIIb) please specify the types of balances (see line 120)
IIIc) give a description of the “Real caloric value of assimilated food”

IV) Statistics
IVa) As you used a replicate you need to include a factor replicate into your statistics; this was not done even if I have asked for it in my first review.
IVb) Describe the multifactorial analysis, be specific and explain the main effects of the post-hoc analysis.

Validity of the findings

The validity of the findings can only be fully explored after we have a concise and specific presentation of the data.

Annotated reviews are not available for download in order to protect the identity of reviewers who chose to remain anonymous.

---

## Round 0.3 · accepted · Accept

Thank you and congratulation for your work revising this manusript. It is now clearly ready for publication.

Reviewer 3 ·

Basic reporting

no comment

Experimental design

no comment

Validity of the findings

no comment

Additional comments

Dear Authors,

I congratulate you on the excellent job to improve the manuscript!

I am happy with the new version and recommend its publication.

Greetings
Anonymous